# A Framework and System Design for Medicines Resources Allocation: A Multi-Stakeholder Assessment of Processes and Electronic Platform Needs

**DOI:** 10.3390/ijerph20053846

**Published:** 2023-02-21

**Authors:** Abdullah Alanazi, Ibtihal Alshatri, Bakheet Aldosari

**Affiliations:** 1Health Informatics Department, King Saud Ibn Abdulaziz University for Health Sciences, Riyadh 11481, Saudi Arabia; 2King Abdullah International Medical Research Center, Riyadh 14611, Saudi Arabia

**Keywords:** medicines, procurement, resources allocation, conceptual framework, system design

## Abstract

Utilizing resources effectively is becoming more critical, especially with the ever-increasing healthcare cost. Little is known about the current practices used by healthcare organizations for the procurement, allocation, and utilization of medical resources. Moreover, the available literature needed to be enriched to bridge the link between resource utilization and allocation processes’ performance and outcomes. This study investigated the processes that major healthcare facilities in Saudi Arabia apply to procure, allocate and utilize medicines resources. The work explored the role of electronic systems and provided a system design and conceptual framework to enhance the availability and utilization of resources. A three-part multi-method, multi-field (healthcare and operational), and multi-level exploratory and descriptive qualitative research design were used to collect the data that was analyzed and interrupted to feed the “future state” model. The findings demonstrated the current state procedure and discussed the challenges and the experts’ opinions on developing the framework. The framework includes various elements and perspectives and is designed based on the results of the first part and was further validated by experts who were optimistic about the inclusiveness of this framework. Some major technical, operational, and human factors were perceived as obstacles by the subjects. Decision-makers can adopt the conceptual framework to gain insights into interrelated objects, entities, and processes. The findings of this study can imply future directions for research and practices.

## 1. Introduction

Building a sustainable healthcare system is a sought-after goal for the viable socioeconomic status of any country [1]. The World Health Organization (WHO) Health System Framework describes the complexity of such a system by providing essential features, including the delivery of good health services; optimally utilized resources to achieve its outcomes; facilitating access to medical products and technologies; sufficient funds and firm leadership and governance to improve health [2]. A well-functioning budgeting and financing strategy is vital in shaping the healthcare system by ensuring access to medicines and vaccines and enhancing quality-of-care outcomes [1,2].

The Saudi Ministry of Health (MoH) is committed to funding, providing services, and overseeing healthcare delivery across different sectors and service levels [3,4]. In 2021, the MoH delivered health services with a budget of over 100 billion SAR [4]. Another report stated that the total pharmaceutical industry expenditure makes up 19.4% of the total health expenditures [5]. Hence, the utilization of resources should be addressed systematically to establish the optimal use of the available resources [6]. Nevertheless, evidence suggests that proper resource allocation in practice is limited [7]. Concerns over some factors, such as poor coordination and communication between different providers and the underutilization of electronic systems, have led to a duplication of efforts and a waste of resources [7,8]. A study indicated 51% of improper utilization by MOH, the private sector, and military hospitals, despite the hospital’s type or ownership [8]. Several studies have discussed the importance of information technology (IT) in enhancing supply chain management processes (SCM) and stated its benefits as improving supply chain agility, higher efficiency, and the ability to deliver resources quickly [9]. However, it is needed to bridge the link between IT utilization and the organization’s performance and outcomes [10]. A top-level integration of multiple sources and stakeholders is required to unleash the full opportunities of such advancements [9]. The key processes and relationships need a consensus among different facilities, and the potential of business process reengineering in healthcare is still overlooked [11]. In addition, the potential of information systems (IS) has yet to be assessed [10].

The presented study aimed to explore and evaluate the current practices of managing medicines and vaccine resources and identify the challenges and opportunities where technology can be employed. Consequently, this study aims to develop a framework to govern the medicine and vaccine utilization processes that decision-makers of such operations could adopt. The system design and the produced framework should aim to integrate the different types of stakeholders who play a critical role in the current state process and explore the future state-version of resource allocation processes. The last objective is to ensure and validate the sustainability of such a framework.

## 2. Materials and Methods

Due to the exploratory nature of this research, a multi-method and multi-field (healthcare and operational), exploratory, and descriptive qualitative research design was used to collect the data that was analyzed and interrupted to feed the “future state” model. Study participants were selected using a purposeful sampling technique. Eight representatives from four stakeholder groups were recruited for this study. Each representative is an expert and is considered a key opinion leader in their field. These individuals provided feedback to describe the current state and shared their opinions regarding the “future state” model. Further, two representatives from the Saudi Food and Drug Authority (SFDA), the main regulatory body for monitoring the drug market, were interviewed. Two representatives from two major public tertiary hospitals and one subject from a private participated in this study. Another representative from the National Unified Procurement Company (NUPCO), a governmental-owned company, was founded to provide unified procurement, logistics, and supply chain services for public healthcare organizations. Another three key opinion leaders were contacted to validate the framework produced after analyzing the results.

### Data Sources and Analysis

The objective and subjective data sources included semi-structured interviews (Appendix A) and secondary data sources provided by the subjects and obtained online. These interviews cover several points including, but not limited to, the current supply chain process, the challenges faced, the main stakeholders of such operation, the followed resource allocation methodology, the outcomes of the existing methods, the feasibility of the electronic supply management system, and the level and extent of user involvement. The interviews were recorded and later transcribed and analyzed using a thematic analysis approach; a sample of the main preset themes built on the objectives of this study and the emergent themes found after analyzing the data provided in the interviews are shown in (Appendix A). Some subjects offered additional secondary data sources, including documents and reports, to enrich the information used to analyze the process.

Next, a conceptual framework was developed based on the findings of the first part of the data collection. Another group of subject matter experts was contacted to evaluate and provide their views on the framework’s validity using a set of questions provided in (Appendix B). To ensure rigor, the semi-structured interviews were conducted face-to-face with subjects in their work environment. Data triangulation and member check techniques were used to enhance the validity and reliability of the data collected. King Abdullah International Medical Research Center (KAIMRC) provided ethical approval to conduct this study (SP18/138/R).

## 3. Results

The findings of this work are presented based on the three stages; the first part describes the results of the exploratory semi-structured interviews and the documents provided by the participants (Table 1). This part is further classified based on the nature of the organization that the participants represented, described in Table 2. The second part expresses the experts’ opinions, from the first phase of the study, about factors that shaped the future state model, presented in (Table 3). The third part contains experts’ opinions regarding the conceptual framework developed in the second phase.

### 3.1. Describing the Current State Process

#### 3.1.1. Regulatory Body Perspective

The Saudi Food and Drug Authority (SFDA), as the main regulatory body for medication product registration, safety evaluation, and monitoring of drugs in the Saudi market, holds no role in procuring medications by healthcare organizations. However, among its many responsibilities, the SFDA oversees monitoring all registered pharmaceutical preparations, its manufacturers, and their sales representatives since they are obligated to ensure the availability of their registered products in the market regardless of their consumption rate or price, according to Article 23 of the Saudi Law of Pharmaceutical Establishments and Preparations [12].

A part of that involves issuing Importation Permits (IP) for medicine products after receiving requests from agents and healthcare facilities. The IP requests include information and justifications related to a healthcare organization’s purchase order (PO) to clear products from the Saudi ports. Currently, the departments of regulatory affairs and importation permits are responsible for reviewing the information and handling the requests submitted using the Importing, Batch-Release and Clearance System (IBRCS). The SFDA reviews the order and will accept or reject the bid within two to three working days based on several factors. It should ensure that its decision does not affect the availability of other drugs in the market. For instance, a request to import a drug product might be rejected if it has registered alternatives. The information provided by the applicant failed to justify the need for requesting the IP, given that healthcare organizations should commit to using products registered to guarantee their safety. This is because of the many steps required to register, test, and evaluate a drug. The process performance measurements for issuing IPs rely mainly on the number of requests and the duration for the review by the staff, and no further analysis is conducted on the data included in these requests.

Interviewees noted that the SFDA does not intervene in areas of resource utilization or allocation since it is the role of the vendors or agents to manage their stock and if they are committed to making the drug available in the market. However, the SFDA responds to complaints made by individuals and private or public healthcare organizations when a drug product is unavailable. Suppose a shortage occurs in a registered product. In that case, the SFDA will contact the responsible person to investigate the reasons behind the shortage, which several reasons, including manufacturing problems or the lack of sufficient demand for a product, can cause. In case of scarcity of non-registered items, the SFDA will issue an importation permit when requested and, in some cases, will contact the agent to discuss solutions to provide and register the drug in the Saudi market. Regardless of the product’s reason or registration status, each shortage incident is assessed and investigated separately to take the proper corrective measures by the parties responsible.

When participants were questioned about the role of technology and information systems, they agreed that the systems used to manage these processes were adequate but could have been more optimal. They helped establish a database and the need to improve some processes. As a result, a new drug track and trace system for pharmaceutical products (RSD) was developed, as part of the National Transformation Program 2020 (NTP), to facilitate the electronic tracking of drug products to monitor their consumption, supply, and safety. A step was preceded by some activities, such as preparing all stakeholders involved, assuring their compliance, and enforcing some measures to ensure that their inputs and outputs are compatible with the new system. One participant emphasized the potential of this system to facilitate the management of drug shortages or overstock cases, providing real-time data about the supply chain and the availability of medications for decision-makers.

Participants expressed similar views regarding the challenges and obstacles that could hinder the SFDA’s role in controlling and governing the market and its related business processes. A significant challenge is a need for more communication and collaboration among the stakeholders, which many tried to overcome by workarounds and solo efforts that led to adverse effects on drug availability in some cases. As well as the lack of integration between the internal and external stakeholders’ electronic systems and shortages in the workforce and resources needed to register and monitor the market caused delays and prolonged regulatory evaluation processes. One participant commented that shortage cases happen in many cases due to problems with the quantification, planning, and forecasting by public and private healthcare organizations.

#### 3.1.2. Client Perspective

##### Public Sector

Two participants from different tertiary hospitals have described fundamentally similar processes in procuring medicine products for their organizations. Mainly, they base their planning on the items of the drug formulary of the hospital, set by the organization’s pharmacy and therapeutics (P&T) committee.

How they acquire drug items follows two tracks. The first is by releases from tenders. The centralized pharmaceutical planning and supply departments estimate quantities, gather the needs, schedule them annually, and then send their estimation to NUPCO. The analysis is built upon collecting information from decentralized planning units across the organizational structure, including the number of consumed quantities, available quantities, and the average monthly consumption rate. The other track is referred to as the direct purchasing process, in which the departments purchase medications, after special requests from the pharmacy department, directly from the vendor or agent.

Consequently, after the organization receives the medications from tender releases, they will check and validate the quantities and the quality of the items and distribute them to departments based on the average monthly usage, and end users in the pharmacy can submit requests for drug items from warehouses. The planning department will evaluate these requests and will either accept or reject these requests. The delivery of these items to the pharmacy inventory is scheduled and dispatched on several factors, including lead time, shelf capacity, and medication shelf life.

Shortage cases are prevalent and usually managed depending on their reason. Several causes can lead to a shortage of a particular drug, such as reasons related to the manufacturing process, delivery failures, or inaccurate planning. Depending on the case’s urgency, the supply department tries to purchase the items directly from vendors or borrow the items from other governmental establishments collaborating in a sharing program. The communication in these cases is carried out using traditional communication channels, e.g., formal letters, not electronically. The issue of overstock has also been discussed. It was mainly managed by announcing the over-stocked items on the organization’s level first and contacting the vendor to deal with the near-expiry items. Finally, by disposing of the items in case no other measures were the case.

The performance measures of these activities contain stock level reports, average consumption rates, and available and consumed amounts. These indicators are drawn from the electronic supply and inventory systems currently used. Which subjects have been perceived valuable in providing live data about the bulk data in warehouses or storage units?

The study participants also discussed some obstacles that may affect this process. It includes issues with the planning stage, where underestimation or overestimation can affect the availability of medications and most frequently contribute to shortage incidents. In addition, redundant work between many governmental departments or internally between departments within the same organization. Not to mention the consequences of some structural changes or strategic planning that might put more stress on could result in strains on the planning departments. Lastly, the limitations are presented with technologies that need more integration between the supply, pharmacy, and warehouse systems. Subjects have expressed concerns about using these systems as a live database of bulk quantities and the problems with manual inventory management.

##### Private Sector

Procuring medicines in the private healthcare sector is similar, in principle, to the public sector. The main difference is that they rely on direct purchases from suppliers since they leverage the flexibility of the private business model and payment terms. Another significant difference is the planning period allocated to procure drugs, they use periods ranging between every month, every four months, and annual periods, and the most used in the representative hospital is the four months for products registered with the SFDA, as opposed to 8-months for the unregistered items. One participant compared that to the more extended period of planning in the public sector, which could contribute to the problem of inaccurate planning.

The representative hospital utilizes a hybrid model, which encompasses the hospital and community pharmacy. So, they base their planning on the actual consumption since their items’ costs are covered by insurance or selling prices, rather than estimating it based on the average monthly. Moreover, their model allows them to request drugs by the brand name approved in their formulary or requested by their pharmacy department. This enables them to negotiate based on quantity to increase their cash flow. The key performance indicators for such a process include: the availability of an item service level or fill rates, forecast accuracy, and expirations of products.

Additionally, the short supply cycles allow them to monitor and replenish their quantiles at a constant rate. As a result, the shortage usually occurs for reasons other than inaccurate planning. Other reasons for the unavailability of certain products were linked to failures in production and delivery and other operational and logistical reasons. However, when a case happens, it is typically dealt with by sourcing for other suppliers or importers and value-to-value exchange activities with other hospitals. Whereas overstock situations, which occur primarily due to over-estimations, are treated by switching products to consume the amounts or contacting the vendor since it is his responsibility to liquidate the stock or withdraw it from the client.

Although these activities need to be fully automated, assistive tools, e.g., Excel sheets and human input are used. One expert expressed his view on the effectiveness of the information technologies by saying it is valuable almost 60% of the time. Since it gives live data about what is “there,” it provides an excellent idea about actual consumption and presents a master list of the products. On the other hand, these systems do not measure the lead time and do not display any triggers for low stocks. The expert related the challenges to the regulations, time-consuming processes, pricing frameworks that could prevent vendors from registering their products in this country, and the uncontrolled shortages due to ungoverned vendor acquisitions.

#### 3.1.3. Service Provider Perspective

The process of procurement for the public sector by NUPCO begins with the clients. All public healthcare organizations will submit their needs within a specified period, following a scheduled plan, to participate in the main tender, which takes place in the first quarter of every year. Afterward, NUPCO will notify all the registered suppliers, using an electronic system, based on their categories to invite them to participate in the tender. Vendors will pay a certain fee to access the template containing information about the items requested in the tenders, approximately around 2000 items in the primary tender, and can bid. Any items not included in the direct tender can be ordered separately in secondary tenders. Then, a specialized team of NUPCO will review the offers made by suppliers and contract with the awarded suppliers. Once the releases to clients are completed, and suppliers can submit their financial claims electronically, NUPCO’s team will validate and checks the orders, fulfill the payments due, close the contract, and evaluate the suppliers for future reference.

Each client is responsible for the planning and estimations of needed quantities. Moreover, NUPCO will evaluate specific performance measures to collect information for strategic purposes, such as lead-time, number of penalties on vendors, number of conflict resolution requests, payments, and the responsiveness of clients and suppliers. Since NUPCO is only responsible for the procurement part of the drug management cycle, it has no role in cases of shortages or overstock.

The electronic systems used to carry out the processes are perceived as very effective by the subject matter expert interviewed for this study. However, it was mentioned that the effectiveness of these systems depends on the level of their efficient utilization by the end user and the level of their involvement. Thus, user resistance was seen as the major challenge from the service provider’s point of view. Other equally important issues included difficulties in performing technical evaluations of requests, the need for highly specialized staff to deal with them, problems that happen because of the ungoverned direct purchase transactions, poor supply planning, and lastly, the lack of downstream and upstream information about the supply chain of medications.

### 3.2. Proposing a New Framework

When participants were questioned about their views on developing a future state framework for procurement and resource allocation and utilization in the Saudi market, they expressed similar opinions reflecting on the challenges they shared with the interviewer (Table 4), regardless of the organization they represent. One participant stated, “We should involve all the stakeholders related to medicines, starting manufacturing and at some stages, even the consumer or patient himself.” They all believe the new framework should account for the changing strategic plans, governmental organizational structures, business models, and the healthcare model. Therefore, it is vital to revise and update the laws, regulations, and guidelines related to some processes, for example, drug registration, product pricing, pharmaceutical planning, and procurement methods. Another equally important concern to consider is to improve the communication channels among the different types of participants to bridge the gaps, encourage active participation, exchange knowledge, and eliminate redundancy and workarounds. Top management commitment is critical, and they should reconsider investing in establishing a unified formulary or drug coding system, further research and development, and prepare highly specialized staff to enhance pharmaceutical planning.

Furthermore, experts highlighted some features to consider when questioning the feasibility of an electronic platform regulating, executing, and monitoring all these related tasks. The majority agreed that these systems should be integrated and interactive to provide maximum benefits in terms of analysis, planning, and forecasting. Old technology, such as barcode and barcode readers, and new technologies, such as predictive models, data mining techniques, and artificial intelligence functionalities, can be utilized to profit from historical and future data. A summary of the suggested solutions by the interviewees is presented in Table 5.

### 3.3. Framework Validation

Subject matter experts perceived this framework as a holistic and inclusive view of all the aspects related to the research questions and objectives. However, they expressed some concerns related to identifying stakeholders, the interdependency between them, and the difficulty in applying these concepts if there was no commitment from decision-makers and higher management. One participant commented that we could use some aspects and focus on the procurement objectives since it would be more applicable.

## 4. Discussion

This work aimed to explore key processes, requirements, objects, and relationships related to medication procurement and resource allocation within the local settings. To develop a model driven by the outcomes of the initial stages of reviewing the existing literature and interviewing a few decision-makers in the Saudi drug market. The findings discussed in the results section provide answers to essential questions to understand the level of availability of medications and who are the main stakeholders, what are the main processes their organizations follow to procure, utilize, and allocate these resources, and explore the challenges they face and provide some suggestions to overcome those obstacles.

### 4.1. The Current State Practices

To begin with the current state process, the study identified the main stakeholders involved in regulating, procuring, utilizing and monitoring medicines in the market. As described previously in Table 3 and the quasi-process in Figure 1, the major players in this process were categorized based on their tasks’ nature and ownership. First is the central regulatory authority, embodied in the SFDA, for the laws, regulations, and guidelines related to the existence and usage of drugs. Their role mainly involved setting the policies for drug registrations and pre-post market surveillance tasks, which entails safety and efficacy evaluations and reviews. It was clear from interviewing representatives from this organization that it does not currently have any role in the process in question. Their involvement appears minor, in principle, and it only revolved around two aspects. First is the clearance process of medications, as all imported drugs must be cleared from the SFDA offices at the designated ports. Therefore, the entity issuing a purchase order for any medication and, by any means of procurement must request a permit. As for the other aspect, the SFDA monitors the availability of drugs in the local market by registering drug products in the local market and responding to claims from healthcare service providers. However, their role appears reactive in this matter. By law, the vendor, manufacturer, or agent is responsible for providing its product in the local market and monitoring its availability [13]. The SFDA is merely making communications using traditional channels to deal with this situation on a case-by-case basis. Although this poses a significant issue, since they cannot validate the information provided by suppliers in case of shortages, the SFDA has recently implemented an electronic tracking tool to aid in providing real-time data, to decision-makers, about the stocks of drugs at each stage of its cycle. The shortcomings in dealing with shortages contributed to the lack of tools and difficulty communicating with other entities, so they were hopeful that the drug track and trace system (RSD) would fill this gap and provide a valuable tool to track and trace the drugs. Since this system is new, there is no way of seeing its impacts on managing drug availability. Moreover, almost all other subject matter experts have emphasized that the laws and regulations of the SFDA need to be revised and updated since many cases of shortage are linked with lengthy registration and evaluation periods from the SFDA.

The second stakeholder group is the private and public sectors and represents the healthcare services providers such as hospitals, clinics, or community pharmacies. Both private and public sectors have established planning and monitoring departments to estimate the different needs, following other methods depending on the tools and practice of rendering healthcare services to patients. The first significant difference observed is the method used to procure medications. The public sector relies mainly on the tenders released by NUPCO, which deals with the suppliers on their behalf. At the same time, the private sector has more flexibility to utilize the direct purchase method since their costs are covered by several means. Another point worth mentioning is the length of the planning period between those sectors; the private sector follows shorter planning cycles allowing them to control their inventory more closely and restock whenever needed. The findings also provide evidence that both entities face the same issues around the lengthy procedures by the SFDA and the poor forecasting functions, which encompasses a need for more trained and qualified workers and problems with integrations.

The third and last stakeholder was described in this study as the service providers. This includes many levels of establishments, such as the semi-governmental company NUPCO and drug manufacturers or the agents importing these products to the market. They are considered the operational side of these processes, unlike the clients who account for the core business aspect. This group’s primary role is to provide the required amounts of drug items to the clients, such as negotiating contracts, closing deals, deliveries and distribution, and warehousing operations.

The findings provide sufficient evidence of the various challenges mainly due to fragmentation in the framework they operate within. However, many initiatives are changing on many levels. The laws and guidelines currently in place are a significant cause of trouble, and a great number of workarounds and repetitions of efforts, which were clearly linked to the poor communication between all the parties involved and can also be related to the lack of awareness of the tasks of each entity. The lack of integration between systems can result from the struggles mentioned above. Not to mention the need for more specialized and knowledgeable workers to operate these functionalities across all levels. Another equally important observation is the lack of reliable performance measurements; the subjects have mentioned many points that were perceived as Key Performance Indicators (KPI). Though the objects mentioned provide data, when put to a real test, it needed to reflect a clear image of what was happening and why we still have utilization issues in our hospitals and the market.

All the views related to the obstacles are well founded, especially when we look at it from each stakeholder’s perspective. Given these points, the idea of establishing a streamlined, easy, and simple way to frame this process with all its related tasks, and incorporate all relevant stakeholders, was appreciated by all the participants. As described in the results section, each has presented their vision about the main points to consider in developing the “future state” model.

### 4.2. The New Conceptual Model

The results of this study have provided us with valuable input in terms of developing a future state framework to enhance and improve the process. A conceptual framework was developed after analyzing the data from the interviews and some other secondary documents. This interdisciplinary framework is not a solution itself. Still, it focuses on a core set of concepts aimed to guide the multiple entities involved to overcome the challenges of the current practices, provide accessible healthcare services, and achieve a sustainable and growing market. The conceptual framework (Table 6) covers four significant activities that could be considered to incorporate all stakeholders and different processes. Each area includes three major concepts that surround the mains objectives of the framework; they are as follows:

Establish:

The Saudi National Formulary (SNF): the SFDA currently issues a national formulary to guide the prescription of registered medicines, including alternative trade names [14]. However, the planning units of clients often need to pay more attention to this list, although it has been proven that adherence to such lists has the potential to reduce medication errors in primary healthcare settings [15]. The framework proposes the establishment of a unified formulary across all public health organizations that addresses the needs of all P&T committees and integrates other factors, such as the priority of locally produced products and the supply chain of drugs. As well as considering the changing model of care, the public sector is moving towards providing national insurance coverage policies. Benchmarking other practices and studies is essential to investigate the impact of adopting a unified national formulary [16].

Office for Harmonization in the Saudi Market (OHSDM): there are several different players and elements to consider in ensuring the availability of medications at reasonable costs. The complexity is seen in their relationships and the potential gaps and areas with an increased rate of conflict between those objects. An office for harmonization should be established to operate under the Ministry of Health’s supervision or on a similar or higher plan to control and maintain the national formulary and oversee the communication and collaboration between different stakeholders. This office should also regulate and provide solutions for non-formulary items and exceptional cases. The main reason for establishing a diverse organization is the need for an independent entity that could provide a framework of cross-national and multi-stakeholder governance.

Research and Development Center (R&D): since the healthcare industry is rapidly evolving, the pharmaceutical industry is definitely catching up. Many changes are happening on global and local scales, whether this includes drug products, supply chain-related business, and the technologies utilized to facilitate these operations [17]. The need for a center for research and development is evident. This need emerges from the need for more qualified workers and contributes to demand and supply functions. As well as the need for local studies and evaluations to map international models and methodologies to our regional settings. Moreover, there is an urgent need for studies to evaluate and analyze the local market needs to produce recommendations based on international standards and best practices locally.

2.Enable:

Product Registration: there was a consensus among the subjects from all categories and the literature that the current law policies and guidelines require a proactive change. In addition, the impact of lengthy processes of product registrations on the availability of medications in the market [18]. Furthermore, improvements in the drug-pricing framework should take place while considering factors such as the quality of products and consumer affordability studies [19]. Unique tracks, expedited processes, and plans should also be implemented to manage urgent cases and critical medications.

Pooled Procurement: the pooled procurement concept entails aggregating a pool of buyers to efficiently group and purchase their demands and offer a positive stand in contractual agreement [20]. The NUPCO should effectively utilize this to replace the traditional bulk purchasing method for the public sector. Another critical point is to study and plan to strategically have centralized and decentralized warehousing and storage options on a cross-national level and utilize electronic platforms to link the service providers and vendors.

Public-Private Partnership: efforts to collaborate and engage with the private sector have been a global trend to address issues related to many [21]. This is a critical aspect to consider as it relates to the objectives of this framework. Efforts should be made to exchange knowledge and experiences with supply chain management practices in different businesses in the private sector. In addition, many non-critical tasks could be outsourced to engage them and share the benefits and risks posed by the market. We should be able to harmonize efforts and align long-term goals and strategies with building a sustainable healthy market.

3.Monitor:

Safety: by reflecting on all the processes, objects, and tasks related to medicines. The monitoring of the safety of pharmaceutical products should be manifested by incorporating regulatory and operational charges and methodologies [22]. For example, we provide means of reporting by effectively utilizing different integrated electronic monitoring tools for supply chain and reporting. As well as the engagement of all stakeholders to ensure constant monitoring and evaluation of drug safety.

Availability: the availability of safe and affordable drug resources is the goal of this model. This can be accomplished by enhancing planning and forecasting tools and utilizing emerging technologies in addition to providing a rich learning environment to improve the quantification and planning techniques and qualified specialists and apply protocols to cycle the available resources within specified frames and guidelines, especially in the public sector, in a way that does not affect the overall availability of the drugs in the marker.

The Utilization Efficiency: the framework emphasizes the monitoring of efficient utilization of resources, whether logistical, operational, or technical. This can be accomplished if we establish a platform or specific programs to share resources between healthcare organizations, we can aim to digitize and encourage participation in borrowing and sharing programs by all stakeholders. Indeed, we can facilitate these objectives by mandating fully integrated pharmacy, health information, and supply chain management systems. Since the technology is available, based on interviewees’ views, it has been proved in the literature that using health information technologies has improved the quality and efficiency of services [22].

4.Support:

Risk Management: failure to notify the regulatory authorities of shortage cases has harmed the management of these cases and the availability of medications on the broad base of clients in the Saudi market [18,22]. Efficient risk prediction and management protocols should be carefully thought out and implemented. Contingency plans from seasonal and sudden shortage incidents should be considered, as well as regulations related to the advance reporting of expected shortages. This will allow access to real-time data for decision-makers and will be reflected in building a solid market.

Localization of Industries: the Saudi vision has carefully presented goals and objectives to change the business environment and improve the overall status of the Saudi market [14]. By encouraging the pharmaceutical establishment, we can contribute to the country’s growth by encouraging investments in the local market and eliminating issues of agents that have highly affected the availability of products in many cases.

Spending Efficiency: improving spending efficiency is another crucial objective of the 2030 vision [21]. Many efforts have been made and still take place to improve administrative policies and procedures. By supporting spending efficiency, we can undoubtedly aim toward sustainable procurement and utilization models and be proactive by establishing reliable and accurate key performance indicator (KPIs).

### 4.3. Implications for Practices

This study was initially motivated by the rising incidents of drug shortages perceived by the author. Apart from gaining an understanding of the current situation and looking at the influencing factors, the proposed framework, or any adaptation that can be derived and tailored to specific needs based on it, is aimed to provide a holistic approach to the procurement and utilization of pharmaceutical, human, and technological resources. The challenges and potential areas for improvements discussed in this study are essential for decision-makers to understand what is happening and gain an insight into what and how we can tackle these issues and offer a return of investment presented in the elevation of the quality of healthcare services and the increased availability and utilization of resources.

### 4.4. Study Limitations

There are a few limitations worth mentioning in this study. First, the common challenge in most qualitative studies is finding an appropriate number of subjects to participate in this research. In addition to the difficulty in obtaining, analyzing, and interpreting processes, challenges form different and conflicting perspectives. Since the information sought required experts’ opinions using a lengthy set of questions, it was not easy to recruit and schedule interviews with key opinion leaders. The second limitation of this study is related to the objects and processes in question. Since it is a very complex set of highly interrelated drug processes, many factors were eliminated during the interviews to keep the focus on the core objectives of the study. Furthermore, the procurement and resource allocation of medicines was assessed through the perspectives of the participated key informants rather than investigating the process itself. Additionally, the study was conducted in Saudi Arabia hence generalizing the results for the procurement and resource allocation of medicines in another context that should be approached cautiously.

## 5. Conclusions

The literature and the findings that founded this study were clear regarding the challenges present in the current practices of the procurement and allocation of resources of medicines. The results demonstrated the type of stakeholders and the current state process. Moreover, it shed light on some of the significant gaps and areas of improvement recognized by all the participants in this study as critical at these changing times. The development of this conceptual framework and its employment during this transitional period of Saudi Arabia considered the major related concepts and affected the process outcomes. It could provide a holistic model to implement at a national level. Accordingly, the implications for practice and the study limitations could provide a future direction since the need for more in-depth research is evident to give a clearer picture of the market needs from an operational point of view and to investigate the high level of complexity and interdependency between the objects of such operations.

## Figures and Tables

**Figure 1 ijerph-20-03846-f001:**
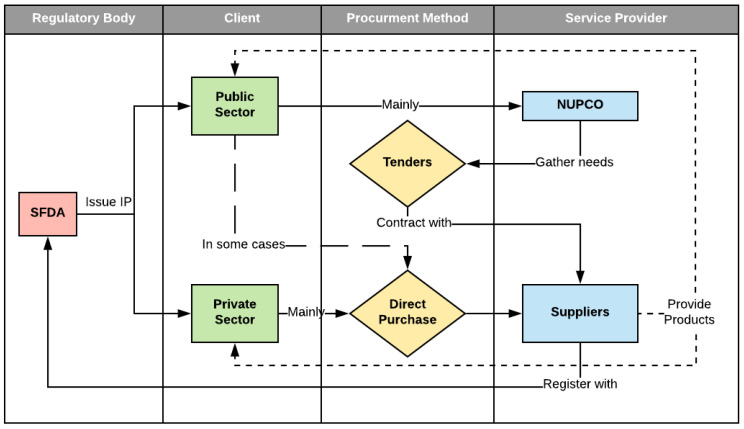
The procurement methods, main stakeholders, and processes.

**Table 1 ijerph-20-03846-t001:** Summary of data sources through system design of the proposed framework.

Stage	Purpose	Data Collection Instrument
1	Investigate current state & future state processes	Semi-structured Interviews Documents and reports
2	Framework Development	Thematic Analysis
3	Framework Validation	Email letters

**Table 2 ijerph-20-03846-t002:** Description of the main stakeholders roles.

Stakeholder	Category	Role
Saudi Food and Drug Authority (SFDA)	Regularity	Drug registration Post-market surveillance Safety & efficacy review and monitoring
Governmental Hospitals	Public Clients	Demand planning Resource allocation Provide healthcare services to patients
Private Hospitals	Private Clients	Procurement and all its related activates Provide healthcare services to patients
National Unified Procurement Company NUPCO	Service Provider	Gather demands Bulk procurement
Suppliers	Service Provider	Provide items Manage shortage/overstock/expired items cases

**Table 3 ijerph-20-03846-t003:** Number of Participants per Category at All Stages.

Subject Category	Number of Participants	Participants Codes
Regulatory Body (MoH + SFDA)	(2 + 2)	P1, P2, P7, P8
Public Clients (Major Public Tertiary Hospitals)	2	P3, P5
Private Clients (Private Hospital)	1	P3
Service Provider (NUPCO)	1	P6
Total	8	

**Table 4 ijerph-20-03846-t004:** Summary of the procurement and resource allocation challenges.

Challenges	Participant Code
Need for revised and updated laws, regulations, and pricing frameworks.	P1, P3, P5, P6, P8
Lack of communication and collaboration between stakeholders.	P1, P2, P3, P5, P7
Issues with redundant work and workarounds.	P1, P5, P6
Lack of integration between systems of pharmacy, supply, warehouses, and other electronic systems/tools.	P1, P2, P3, P4, P5, P6
Poor quantification, planning and forecasting.	P1, P2, P3, P4, P5, P6
Shortages in qualified and trained supply specialists.	P4, P5, P6
Prolonged drug registration and analysis processes.	P3, P4, P5, P8

**Table 5 ijerph-20-03846-t005:** Summary of the suggested solutions and Priorities for the Future State Framework.

Top Priorities	Suggested Solution
Track, trace, supply chain, inventory management tools	Utilize electronic systems Integration across systems
Proper planning and quantification	Planning and quantification training Integration between systems Skilled and trained personnel
Laws and regulations	Update and revise Communication with all stakeholders
Accelerated processes	Collaboration with all stakeholders Outsourcing
Research and development	Specialized centers/units Training Local market analysis Adopt best international practices

**Table 6 ijerph-20-03846-t006:** The conceptual framework for the procurement and resources allocation of medicines.

Establish	Saudi National Formulary (SNF)	Office of Harmonization in Saudi Drug Market (OHSDM)	Research and Development Center (R&D)
Enable	Product registration	Pooled procurement	Public Private Partnership (PPP)
Monitor	Safety	Availability	Utilization efficiency
Support	Risk management	Localization of industries	Spending efficiency

## Data Availability

Not applicable.

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
