# Peer review of "A Framework and System Design for Medicines Resources Allocation: A Multi-Stakeholder Assessment of Processes and Electronic Platform Needs"

_ijerph, 2023, doi:10.3390/ijerph20053846_

Round 1

Reviewer 1 Report

Manuscript is interesting for health care management and should be useful. 

The main obstracles that all story was dedicated for Saudi practice and state of the art, only from publication that discuss the national situation. The more wither disscusion is need for the atractk the scientific audience!!!

One of global/international phenomenon has now medicines shortage, and this is obligatory to include as approach regarding the theme of scientific observation..

The graphic presentation Fig ! for the offered Framework, should be more precise.

Some comment/corrections was include in attach pdf document.

Author Response

I would like to express my gratitude for taking the time to review our work.

Reviewer 2 Report

This manuscript is a very methodical and comprehensive study regarding the assessment of a national resource allocation for medications and with recommendations how to improve the current system. The article is well written and well organized. This study nicely integrates macro policy levels with business policies at the organizational and stakeholder level.

I have several comments, which may help focus the main points of the article. 

General Comments:

  1. It would be helpful to state early in the article that the study involves the Ministry of Health for Saudi Arabia. This is not clearly stated.

  2. The article refers to “ as-is” and “ to-be”. It would be helpful if you refer to this as “current state” and “future state”. Which is considered the typical terminology in process engineering.

  3. The article refers to a Framework, which is appropriate but it would be helpful to incorporate the concept of system design into the article as well as the title.

Specific Comments:

  1. Line 68- Indicates that there are six representatives for this study, but Table 1 represents that there are eight participants in this study.

  2. Line 72- Refers to SFDA. Please identify what this means.

  3. Table 2- The third column is labeled Data Collection Instrument; however, stage two indicated it is analysis and not analytic (graphs). 

  4. Please spell out or put in footnote SFDA and NUPCO.

  5. Line 112- 3.1 should be described as the “Current State Process”.

  6. Line 283- 3.2 This section refers to a new framework and appropriately includes a quasi process in Figure 2.0.

  7. Line 402- Describes a new conceptual model which could also be described as a “Future State Process”.

  8. Figure 2.0- Again, spell out or footnote SFDA and NUPCO 

  9. Line 508- Define KPI

  10. Line 511- Refers to one author. These should be authors.

  11. Section 4.4 (520-529)- The discussion predominantly refers to the difficulties of the study. However, the limitations are not clearly identified. Also, you did not actually study the process; rather key informants were asked about the process. This distinction should be made in the limitations section.

Author Response

(The authors gave the same response as above.)
